# Whole-Genome Sequencing and Genetic Diversity of Human Respiratory Syncytial Virus in Patients with Influenza-like Illness in Sicily (Italy) from 2017 to 2023

**DOI:** 10.3390/v16060851

**Published:** 2024-05-26

**Authors:** Fabio Tramuto, Carmelo Massimo Maida, Giulia Randazzo, Valeria Guzzetta, Arianna Santino, Rita Li Muli, Claudio Costantino, Giorgio Graziano, Emanuele Amodio, Walter Mazzucco, Francesco Vitale

**Affiliations:** 1Department of Health Promotion Sciences Maternal and Infant Care, Internal Medicine and Medical Specialties “G. D’Alessandro”—Hygiene Section, University of Palermo, 90133 Palermo, Italy; carmelo.maida@unipa.it (C.M.M.); claudio.costantino01@unipa.it (C.C.); emanuele.amodio@unipa.it (E.A.); walter.mazzucco@unipa.it (W.M.); francesco.vitale@unipa.it (F.V.); 2Regional Reference Laboratory for Molecular Surveillance of Influenza, Clinical Epidemiology Unit, University Hospital “Paolo Giaccone”, 90133 Palermo, Italy; giulia.randazzo1986@gmail.com (G.R.); valeria.guzzetta@policlinico.pa.it (V.G.); arianna.santino@policlinico.pa.it (A.S.); rita.limuli@policlinico.pa.it (R.L.M.); giorgio.graziano@policlinico.pa.it (G.G.)

**Keywords:** respiratory syncytial virus, whole-genome sequencing, molecular surveillance, amino acid change, mAb, vaccine, community, Sicily, Italy

## Abstract

Monitoring the genetic variability of human respiratory syncytial virus (hRSV) is of paramount importance, especially for the potential implication of key antigenic mutations on the emergence of immune escape variants. Thus, to describe the genetic diversity and evolutionary dynamics of hRSV circulating in Sicily (Italy), a total of 153 hRSV whole-genome sequences collected from 770 hRSV-positive subjects between 2017 and 2023, before the introduction of expanded immunization programs into the population, were investigated. The phylogenetic analyses indicated that the genotypes GA.2.3.5 (ON1) for hRSV-A and GB.5.0.5a (BA9) for hRSV-B co-circulated in our region. Amino acid (AA) substitutions in the surface and internal proteins were evaluated, including the F protein antigenic sites, as the major targets of immunoprophylactic monoclonal antibodies and vaccines. Overall, the proportion of AA changes ranged between 1.5% and 22.6% among hRSV-A, whereas hRSV-B varied in the range 0.8–16.9%; the latter was more polymorphic than hRSV-A within the key antigenic sites. No AA substitutions were found at site III of both subgroups. Although several non-synonymous mutations were found, none of the polymorphisms known to potentially affect the efficacy of current preventive measures were documented. These findings provide new insights into the global hRSV molecular epidemiology and highlight the importance of defining a baseline genomic picture to monitor for future changes that might be induced by the selective pressures of immunological preventive measures, which will soon become widely available.

## 1. Introduction

Human respiratory syncytial virus (hRSV) is the leading cause of hospitalizations due to severe acute respiratory tract infections, bronchiolitis and pneumonia in infants and children.

In low- and middle-income countries, the pathogen is the most common cause of childhood lower respiratory tract infections (LRTIs), with about 3.2 million hospitalizations and over 118,000 deaths worldwide [1].

Although it is estimated that hRSV affects almost all children worldwide at least once within the first two years of life [2], the global burden of hRSV infections extends beyond very young children, remaining frequent at later ages also [3,4], including subjects with underlying medical conditions [5,6], the elderly [7,8] and immunocompromised individuals [9], who are increasingly being recognized as vulnerable populations at significant risk of severe infection and death.

In addition, reinfections may occur throughout life because of the lack of complete or long-lasting immunity, which is usually associated with milder outcomes in older children and healthy adults.

hRSV is a single-stranded, negative-sense RNA-enveloped virus with a genome, approximately 15 kb long, containing ten genes that encode eleven different proteins; three of those are non-structural proteins and eight are structural proteins, either internal or external [10].

Among these, the two surface glycoproteins, the attachment (G) and the fusion (F), are fundamental to promote viral infectivity and pathogenesis, being strictly required to mediate virus entry.

The G protein is much less conserved than F protein. Based on the antigenic and genetic variations mainly associated with distal third of the G gene, the second hypervariable region (HVR2) [11], hRSV is classified into two subgroups, hRSV-A and hRSV-B, which can co-circulate within a community during a seasonal outbreak, though usually one subgroup dominates the other [3].

Due to its variability, HVR2 has been historically used to investigate the molecular epidemiology of hRSV and to further categorize viral genomes into various genotypes [12].

At present, the two genotypes ON1 (hRSV-A) and BA9 (hRSV-B) are dominant globally [13], replacing all previous variants of hRSV, suggesting an evolutionary advantage to these genotypes. Interestingly, both ON1 and BA9 harbor a 72- or 60-nucleotide duplicate insertion in HVR2 of the G gene, respectively [14], probably representing a genetic marker of immune escape at a population level [14,15].

The F protein is synthesized as an inactive precursor (F0) that undergoes cleavage by host cell proteases, generating two major subunits, F2 and F1, and resulting in the removal of an internal glycopeptide of 27 amino acids, named p27 [16,17].

Moreover, during this process, the F protein readily modifies its structure, passing from a metastable conformation, named pre-fusion, that is incorporated into the virus particle, to a lower-energy, highly stable post-fusion conformation, refolded during viral fusion to the cell membrane [10].

In contrast to the G protein, the F protein is known to have a high level of sequence conservation among hRSV subgroups/genotypes, is more immunogenic than glycoprotein G, and also contains six epitopes (Ø-V) capable of inducing a neutralizing antibody response. While sites I, II, III and IV are found in both fusion conformations [10], sites Ø and V are lost during the rearrangement of the F protein. Importantly, these latter ones are able to bind monoclonal antibodies (mAbs) with a stronger neutralization capacity.

Therefore, most of treatments available for the prevention of hRSV disease based on prophylactic mAbs [18] or licensed vaccines, which are next to be released or in the pipeline [19], target the pre-fusion glycoprotein F.

In particular, the first prophylactic mAb introduced to prevent hRSV infection, palivizumab, targets the antigenic site II of the F protein, while the next-generation mAb, nirsevimab, binds the highly sensitive antigenic site Ø, as is the case for most candidate or recently approved vaccines [10]. Finally, in addition to nirsevimab, a second next-generation mAb actually in clinical phase 3, clesrovimab, is directed against the F protein antigenic site IV [20].

To date, a unique vaccine (Abrysvo™) has been approved for use in pregnancy, with the aim to passively protect infants against LRTI within the first semester of life [21], while two different vaccines (Abrysvo™ and Arexvy™) have been licensed to protect adults ≥60 years of age [22].

In addition, an mRNA-based vaccine will be introduced shortly for use in adults 60 years of age and older [23]. All these vaccines target the pre-fusion conformation of the F protein.

However, besides F and G glycoproteins, there is also interest in targeting other hRSV surface glycoproteins in the development of vaccines, such as the small hydrophobic protein [16,24,25] as well as the highly conserved nucleoprotein (N), the transcription elongation factor (M2-1) [26,27,28,29] or any of the other internal proteins [10,20].

Furthermore, recent studies have also indicated the p27 peptide of hRSV F protein as an immunodominant antigen in young children, eliciting a protective immune response comparable to other known hRSV site-specific neutralizing antibodies [30] by inducing an antibody-dependent cell cytotoxicity (ADCC) and T cell-mediated effector functions rather than a neutralizing antibody activity [24,31].

Notably, surveillance studies on the molecular variability of hRSV subgroups have clearly demonstrated how this virus can evolve over time, exhibiting several sequence polymorphisms, each of which may occur at different evolutionary rates.

In this regard, Mas et al. [32] found that the antigenic sites III and IV were the most conserved regions of the F protein, as also described by Hause et al. [33] for the antigenic site IV across all genotypes.

Considering that a single amino acid change may also have a high impact on antibody binding, these antigenic sites have been proposed as more desirable targets, carrying a reduced risk of generating immune escape viruses [20].

Nevertheless, although the F protein is generally well conserved, a certain degree of variability has been observed in the signal peptide, transmembrane domain and antigenic sites [17,32], derived from viral evolution due to the combined effect of error-prone RNA polymerase, genetic drift and immune pressure [34].

In this regard, a promising mAb in clinical phase 3 targeting the site V, suptavumab, was recently discontinued from further development because of an almost complete lack of neutralizing activity sustained by the emergence of two spontaneous mutations in the F protein of circulating hRSV-B strains (L172Q and S173L) [35].

For these reasons, as seen with influenza, continued molecular surveillance of hRSV is essential to investigate the epidemiology of the virus and the circulation of subgroups; it is also an invaluable tool for understanding the potential underlying effect of immune pressure on the genetic diversity and polymorphisms in circulating viruses, not only in the F gene but over the entire genome, potentially affecting the overall performance of therapeutics and vaccines [36,37,38].

This study is the first genome-scale analysis on hRSV epidemiology in Sicily (Italy) and aimed to fill the information gap still present in our country, examining the sequence variability of hRSV whole-genome sequences obtained between 2017 and 2023.

## 2. Materials and Methods

### 2.1. Study Population and Case Definition

Between October 2017 and April 2023, oro-pharyngeal swabs were collected through the running surveillance program in Sicily, the fifth most populous region of Italy, within the national network RespiVirNet (formerly known as InfluNet; https://respivirnet.iss.it/default.aspx, accessed on 26 May 2024) from patients of any age who met the case definition of ILI (influenza-like illness) or SARI (severe acute respiratory infection) [39,40,41].

In particular, a case of ILI was defined as a person presenting a sudden and rapid onset of at least one of the systemic symptoms among fever or feverishness, malaise, headache, or myalgia and at least one of the respiratory symptoms among cough, sore throat, or shortness of breath. A case of SARI was defined as a patient with an acute respiratory infection that required hospitalization. All specimens were conferred to the Sicilian Regional Reference Laboratory of the RespiVirNet network, operating at the University Hospital “AOUP P. Giaccone” of Palermo (Italy), and stored at −80 °C until further use.

The determination of the hRSV subgroup, amplicon-based whole-genome next-generation sequencing (NGS) and assembly were conducted as follows:

Viral nucleic acids were extracted from specimens using a QIAmp Viral RNA extraction kit (QIAGEN, Hilden, Germany) according to the manufacturer’s instructions. Each sample was tested for hRSV-A and hRSV-B using singleplex one-step real-time (rt) retrotranscription (RT) assays [3,42]. A test was considered positive when its cycle threshold (Ct) value was <40.

For both hRSV-A and hRSV-B, the reactions were performed in a final volume of 15 μL, including 5 μL of template, 3.9 μL of QuantiNova Pathogen mastermix (QIAGEN, Germany), 0.15 μL of 30 μM forward primer, 0.15 μL of 30 μM reverse primer, 0.3 μL of 10 μM probe and 5.5 μL of nuclease-free water.

The thermal profile was as follows: reverse transcription at 50 °C for 15 min, followed by 2 min at 95 °C and 45 cycles of 95 °C for 15 s and 55 °C for 30 s.

All rt-RT-PCR assays were performed with a QuantStudio^TM^ 7 Flex Real-Time PCR System (Applied Biosystems, Waltham, MA, USA).

Each hRSV-positive sample with an rt-RT-PCR Ct value ≤ 30 was considered suitable for whole-genome sequencing (WGS) of hRSV. To this purpose, a multi-segment PCR pre-enrichment approach was used to sequence both hRSV-A and hRSV-B viral genomes, according to the schemes depicted in Appendix A.

In short, RT-PCR was used to amplify four overlapping amplicons, each covering about 4000 nucleotides that together span the entire hRSV genome.

Each RT-PCR reaction was performed using the SuperScript™ IV One-Step RT-PCR System with ezDNase (Invitrogen, Waltham, MA, USA) in a final volume of 50 μL, including 25 μL of Platinum SuperFi RT-PCR Master Mix, 0.5 μL of SuperScript IV RT Mix, 2.5 μL of 10 μM forward primer, 2.5 μL of 10 μM reverse primer, 15 μL of template pre-treated with ezDNase and 4.5 μL of nuclease-free water.

The thermal profile was as follows: reverse transcription at 55 °C for 10 min, 2 min at 98 °C and 40 cycles of 98 °C for 10 s, 61 °C per 10 sec and 72 °C for 3 min, followed by a final extension of 5 min at 72 °C. All primers are listed in Appendix A.

The size and yield of each RT-PCR was determined with the Agilent Tapestation; then, equimolar amounts of each of the four amplicons were pooled.

The pooled hRSV amplicons were then purified with Agencourt AMpure XP beads (Beckman Coulter, Brea, CA, USA), quantified using the Qubit dsDNA quantitation HS assay kit (Invitrogen, USA) and then prepared for sequencing with Ion Xpress Plus Fragment Library Kit (Thermofisher, Waltham, MA, USA), following the manufacturer’s protocol. Barcoded libraries of 200–300 bp were sequenced on an Ion GeneStudio S5 Prime System (IonTorrent, Thermofisher).

The coverage analysis, read mapping and assembly of the sequencing reads into target whole-genome nucleotide consensus sequences were performed using CLC Genomics Workbench version 23.0.5 (QIAGEN Digital Insights, Aarhus, Denmark). All hRSV sequences were deposited on the Global Initiative on Sharing Avian Influenza Data (GISAID, Munich, Germany) database (EPI_ISL_19063348 to EPI_ISL_19063420 for hRSV-A; EPI_ISL_19063268 to EPI_ISL_19063347 for hRSV-B).

### 2.2. Phylogenetic Analysis

Preliminary analyses were carried out by using Nextclade tool version 3.2.0, available online (https://clades.nextstrain.org, accessed on 26 May 2024), to assign the clade to each Sicilian hRSV sequence [43] and to explore the most probable phylogenetic placement. Therefore, two different subsets of hRSV-A and -B sequences collected worldwide were built and used as reference viral strains in the construction of phylogenetic trees, in order to place our datasets into a global context. Sequences without a geographic association or sampling date were excluded.

To this purpose, each sequence dataset was aligned with MAFFT version 7, available online (https://mafft.cbrc.jp/alignment/server/index.html, accessed on 26 May 2024), and the neighbor-joining method implemented in the MEGA X package was used for reconstructing phylogenetic trees from evolutionary distance data [44]. The reliability of the tree topology was estimated by using the bootstrap re-sampling method with 1000 replicates. The best-fit evolutionary model and parameters were calculated and the Tamura–Nei model of nucleotide substitution (TN93 + G + I) was estimated as the most appropriate for the datasets.

### 2.3. Analysis of Deduced Amino Acid Sequences and Mutations

Deduced amino acid (AA) sequences were predicted for both subgroups with standard genetic code and were aligned to their respective reference sequences (GenBank accession number: KT992094 for hRSV-A and AY353550 for hRSV-B). For each of the eleven viral proteins, the AA positions that differed from the corresponding prototype strain were defined as mutated. AA variability was reported as the percentage of occurrence of each mutated AA at a given residue position found in a subgroup.

### 2.4. Selection Pressure Prediction

The ratio of non-synonymous/synonymous substitutions (dN/dS) was considered when evaluating codons under selective pressure. dN/dS were analyzed using three different methods: the fixed effects likelihood (FEL), the mixed-effects model of evolution (MEME) and the fast unconstrained Bayesian approximation (FUBAR) methods. All algorithms were implemented in the Datamonkey webserver (https://www.datamonkey.org, accessed on 26 May 2024). A positively selected residue was considered significant at a *p*-value threshold = 0.1 or posterior probability = 0.9.

### 2.5. Statistical Analysis

The study population was arbitrarily subdivided into nine different age groups: five for children/teenagers (≤11 months, 12–23 months, 2–4, 5–10 and 11–18 years old) and four for adults/elderly (19–34, 35–49, 50–64 and ≥65 years old).

Descriptive statistics were used to summarize the socio-demographic and clinical data of the recruited patients, as well as for viral characteristics. Frequencies and percentages were reported for categorical variables.

The data were processed with the STATA MP statistical software package v16.1 for Apple^TM^ (StataCorp LLC, College Station, TX, USA).

### 2.6. Ethical Review

This study was carried out in full compliance with the rules concerning the protection of personal data adopted in Italy and in accordance with the Helsinki Declaration. The participants were tested for hRSV as part of clinical management and for surveillance purpose. All data were analyzed anonymously. Therefore, additional informed consent for this laboratory-based study was not required.

The study was approved by the institutional ethics committee of the University Hospital “AOUP P. Giaccone” of Palermo (Italy), approval number 09/2019, and was partially funded by Merck Sharp & Dohme Corp (Rahway, NJ, USA). The sponsor of the study had no role in the study design, data collection, data analysis, data interpretation or writing of the report.

## 3. Results

### 3.1. Demographic Characteristics of Study Population

Table 1 describes the demographic characteristics of the hRSV-positive subjects included in the study. A total of 13,193 oro-pharyngeal swab specimens were collected in Sicily from October 2017 to April 2023, of which 770 (5.8%) tested positive for hRSV using one-step real-time PCR. Among these, 335 (46.1%) were hRSV-A and 401 (52.1%) were hRSV-B, whereas 14 (1.8%) were co-infected with both subgroups.

Overall, hRSV was found in all age groups considered. Similar proportions of hRSV-A and hRSV-B were observed among subjects ≤ 34 years old, while hRSV-B significantly prevailed over hRSV-A among adults and the elderly. Co-infections were documented only among children ≤ 10 years of age.

The male-to-female ratio was 1.02, and also, no differences were found in terms of proportions of hRSV subgroups according to sex.

The study population was mostly community-based for 92.6% (*n* = 713/770) of total subjects, among which the detection of hRSV subgroups was similarly represented (47.7% vs. 50.3% for hRSV-A and hRSV-B, respectively). Instead, hospital-based infections were predominantly sustained by hRSV-B (73.7%, *n* = 42/57).

Both viral subgroups co-circulated during the six surveillance seasons analyzed, showing the pattern “BBABBB”, even considering the extremely low frequency of hRSV occurrence during the 2020–2021 season.

### 3.2. Phylogenetic Analysis of hRSV Genomes

In total, 153 high quality whole-genome sequences were obtained, 73 (47.7%) hRSV-A and 80 (52.3%) hRSV-B, proportionally distributed across seasons (Table 2).

Preliminary phylogenetic analyses were carried out with the aim of highlighting the main clusters of evolution and potentially divergent genomes, as well as to obtain the clade/lineage assignment.

Overall, several clades and subclades were identified among hRSV-A strains, designated as A.D (*n* = 9), A.D.1 (*n* = 16), A.D.2.1 (*n* = 1), A.D.2.2 (*n* = 19), A.D.3 (*n* = 6), A.D.3.1 (*n* = 6), A.D.5 (*n* = 7) and A.D.5.2 (*n* = 9), grouping into five different clusters, with all belonging to the genotype GA.2.3.5 (equivalent to the genotype ON1) (Figure 1).

On the other hand, as depicted in Figure 2, hRSV-B genomes were classified as B.D.4.1 (*n* = 4), B.D.4.1.1 (*n* = 52) and B.D.E.1 (*n* = 24). These were aggregated in two main evolutive clusters, with the exception of a small number of ungrouped sequences. Altogether, hRSV-B strains belonged to the genotype GB.5.0.5.a (equivalent to BA9).

Based on these preliminary analyses, phylogenetic reconstructions of simplified datasets were built in order to point out potential evolutionary patterns or seasonal clusters.

In particular, as illustrated in Figure 3, the clades A.D, A.D.2.1, A.D.2.2 and A.D.5 were mainly represented by hRSV-A strains that circulated between 2017 and 2018 and 2019 and 2020, whereas genomes obtained in more recent seasons (2021–2022 and 2022–2023) predominantly belonged to the clades A.D.1, A.D.3.1 and A.D.5.2 and were temporally consistent with other strains collected worldwide.

Overall, the phylogenetic relationships observed among hRSV-A genomes suggested multiple introductions of this subgroup in Sicily.

Similarly, although ascribing to a fewer number of clades, the NJ tree topology of hRSV-B genomes highlighted seasonal patterns of different lineages within both B.D.E.1 and B.D.4.1.1 clades. The latter included sequences collected over a long period covering the seasons 2017–2018 and 2021–2022, whereas a small number of sequences more recently collected between 2021 and 2022 and 2022 and 2023 fell into B.D.E.1 (Figure 4). Finally, the clade B.D.4.1 grouped only four genomes of viruses circulating early in 2017–2018 and 2018–2019.

### 3.3. Amino Acid Polymorphisms in hRSV F Surface Proteins

Figure 5 illustrates the AA variation frequencies detected in the F protein open reading frame.

Overall, 9.6% (*n* = 55/574) of AA sites were polymorphic, with frequencies ranging from 1.2% to 100.0%. Of these, 20.0% (*n* = 11/55) were in the signal peptide (SP) region, 21.8% (*n* = 12/55) in the F2 chain, 12.7% (*n* = 7/55) in the p27, 36.4% (*n* = 20/55) in the F1 chain and 9.1% (*n* = 5/55) in the region including the transmembrane domain (TM) and the cytoplasmic tail (CT).

In total, hRSV-A strains showed a higher number of mutated sites (6.3%, *n* = 36/574) when compared to hRSV-B (4.5%, *n* = 26/574), although hRSV-B strains had more AA changes than hRSV-A within the key antigenic sites (Table 3 and Figure 5). More in detail, among hRSV-A, the most frequently occurring mutations were K66E (100.0%) at the antigenic site Ø, V384I (100.0%) at antigenic site I, N276S (90.4%) at antigenic site II and V152I (100.0%) at antigenic site V. Some other AA changes were barely represented (1.4%), such as K65R and I57V at sites Ø and V, respectively.

On the other hand, among hRSV-B strains, more non-synonymous substitutions were found at a higher mutation rate, such as S197N, I206M and Q209R (100.0%, 93.7% and 92.5%, respectively) at the antigenic site Ø, F45L (100.0%) at the antigenic site I and K191R (93.8%) at the antigenic site V.

Moreover, S389F/P at site I and S190N at site V were detected in about one-third (35.0% and 31.2%, respectively) of Sicilian strains, while R42K at site I, S276N at site II and E463D at site IV were, respectively, found in 3.8%, 2.5% and 1.2% strains. Notably, no AA variations were detected at site III of both subgroups and at site IV of hRSV-A, whereas a change at residue 276 was the only shared between subgroups, and each subgroup had a preferential amino acid.

Finally, some AA mutations were also identified in the internal glycopeptide p27, two of which were A122T and K124N, which are particularly frequent among hRSV-A strains (78.1% and 100.0%, respectively).

### 3.4. Determination of the Selection Pressure on hRSV F Gene

Amino acid changes on the F protein may result from selective pressure being exerted on the virus by the host’s immune response during infection. The potential positive selection of these residues was determined by examining their rate of change (dN/dS) (Appendix A).

The site-by-site selection analysis showed that both negative and positive selection happened in the F gene, with 43 and 35 sites that were, respectively, defined as negative selection in RSV-A and RSV-B, while two residues at F protein position 122 in hRSV-A and 125 in hRSV-B circulating strains were defined as being under positive selection (using at least two of the three methods).

### 3.5. Analysis of NS1, NS2, N, P, SH, M2-1, M2-2 and L Proteins

For the sake of completeness, each of the other hRSV proteins were evaluated with the exception for the G protein, which was already discussed by our research group [45].

Appendix A illustrate the distribution of mutated AA positions identified in both structural and non-structural proteins, some of which were highly recurrent and, on average, more commonly documented among hRSV-B than hRSV-A.

In more detail, as compared to the prototype strains A2 and 9320, respectively used in this study of hRSV-A and hRSV-B, the percentage of changed AA residues in the non-structural protein 1 (NS1) of both subgroups was 5.8%, while the non-structural protein 2 (NS2) varied in 6.5% of hRSV-A and 11.3% of hRSV-B isolates.

In the other structural proteins, including the nucleoprotein, the phosphoprotein (P), the matrix protein (M), the transcription processivity factor M2-1 and the large polymerase complex (L), we found significantly lower mutation rates ranging from 0.8% to 6.1%. Of note, the small hydrophobic protein (SH) harbored 10.8% of the changed AA positions among hRSV-A and 16.9% among hRSV-B.

Finally, the non-structural M2-2 protein exhibited the highest proportion of mutated amino acids, reaching the percentage of 22.6% in hRSV-A and 15.0% in hRSV-B strains.

## 4. Discussion

This study investigated the genetic heterogeneity of hRSV whole-genome sequences, collected in Sicily (Italy) during six consecutive cold seasons from 2017 to 2018 and 2022 to 2023, among patients presenting ILI symptoms in order to provide insight into the molecular epidemiology of this pathogen and the background AA variability before the introduction of expanded immunization programs into the population.

As already described in a previous report by our group [3] and in agreement with other authors [1,7,46], most of the hRSV-positive subjects were infants and young children, even though hRSV was detected among individuals of any age, including the elderly.

Viral subgroups showed dynamic epidemiological patterns in which one subgroup prevailed over the other from year to year. Consistent with other reports [47,48], the pattern “BBABBB” was observed in Sicily. We found that GA2.3.5 and GB5.0.5a (equivalent to ON1 and BA9, respectively) were the predominant genotypes; these substantially co-circulated, though the relative proportions varied seasonally.

On a global scale, after the first discovery of ON1 in 2010 [49] and BA9 in 2005–2006 [50,51], the two genotypes have completely replaced all previous variants, becoming dominant worldwide [52,53,54,55,56,57,58,59,60,61,62].

Through the phylogenetic analysis of whole genomes, we found that hRSV-A and hRSV-B were closely related to strains circulating in several countries. In general, the tree topologies suggested multiple entries of both hRSV subgroups in Sicily, with dominant lineages varying over time, often coherently with each seasonal outbreak or related to local clusters of viral spread. This observation partially agreed with the results reported by Lai and colleagues [47], who documented, in Northern Italy, a less pronounced genetic diversity, even considering that the study referred to a pediatric cohort of the single epidemic season 2021–2022.

In Sicily, the molecular epidemiology of hRSV during pre- and post-COVID-19 surveillance seasons has been described [45], though this study focused on the evolutionary analysis of the G protein gene only.

However, strategies currently adopted in the active and passive primary prevention of hRSV infections target different surface glycoproteins. Among them, the F protein demonstrated the most promising results because of its unique properties involving its role in cell entry, its higher genetic conservation and the presence of antigenic epitopes in the pre-fusion conformation, where antibodies may bind with the strongest neutralizing activity [20].

Because of their quasispecies nature, RNA viruses, including hRSV, undergo constant evolution. Therefore, immune escape mutants may emerge as a consequence of adaptation to changing selective pressure [63], also potentially driven by active and passive immunization campaigns [64,65].

Therefore, although F protein is relatively less variable than other proteins among different subgroups and genotypes, several AA variations have been documented in this genetic region, which might lead to some differences in the efficiency of the host immune response.

Consequently, it is crucial to investigate F protein sequence evolution and AA mutations in contemporary circulating strains, which may potentially affect the performance of mAbs and vaccines.

Overall, the diversity of Sicilian hRSV strains was relatively low, with about 10% of mutated AA residues distributed throughout the entire open reading frame of the F protein, essentially documented as spontaneous polymorphisms due to the natural evolution, since an immunization program has not yet been implemented in Sicily. According to other reports [34,56,66], viruses belonging to the subgroup A showed a higher number of mutated sites than B, although hRSV-B strains had more variability within the antigenic sites Ø-V and p27. The hRSV-A F protein showed six AA substitutions at antigenic sites Ø, I, II and V, four of which occurred at a very high frequency (>90%) compared with the reference strain. The hRSV-B strains had ten non-synonymous AA changes distributed over five antigenic sites (Ø, I, II, IV and V), with six AA changes highly recurrent.

In particular, K66E/V384I/N276S/V152I (at sites Ø, I, II and V, respectively) of hRSV-A were also observed at similar frequencies by Sun et al. [56], as well as S197N/I206M/Q209R at site Ø, S276N at site II and K191R at site V among hRSV-B strains [56].

Furthermore, the three AA substitutions I206M, Q209R and K191R, earlier documented in the USA and Europe [55,66] in most of hRSV-B strains collected between 2018 and 2023, were found at a low frequency in South Africa from 2015 to 2017 [34], thus suggesting an important increase in these AA changes in more recent times only.

It is interesting to note that the K65R found in hRSV-A or the mutations I206M/Q209R documented in hRSV-B do not appear to significantly reduce the neutralization activity of mAbs binding the antigenic site Ø, such as nirsevimab [36].

Nevertheless, Adams et al. [67] reported an hRSV-A variant carrying the mutation N276S located at site II, which was predominant in Sicily as well as in Canada [68], potentially conferring partial resistance to palivizumab and able to promote the selection of a secondary AA residue variation K272E, leading to complete resistance. Conversely, Zhu and colleagues, in 2012 [69], assessed that isolates expressing N276S in hRSV-A or S276N in hRSV-B did not affect susceptibility to this mAb.

The development of resistance to mAbs should be considered a natural consequence of rapid replication of the virus in the presence of selective pressure, and a widespread use of mAbs against hRSV for all infants may represent a potential risk of selecting immune escape viruses. Additionally, it has been reported that subpopulations of hRSV may develop some natural polymorphisms potentially associated with the resistance to antiviral agents in vitro, which do not necessarily lead to the clinical failure of the treatment in vivo.

This is the case for palivizumab, which is able to bind the antigenic site II on both the pre-fusion and post-fusion forms of the hRSV F protein, for which single AA substitutions such as N262D/S, N268I, K272N/M/T/Q and S275F/S/L or any combination thereof may provide neutralizing resistance in cell culture or animal models [69,70,71,72,73], although the clinical failure of prophylaxis because of mAb-resistant viruses has not been reported to date in post-marketing analyses [36].

In addition, the internal glycopeptide p27 in the F protein has been demonstrated to stimulate a strong immune response in hRSV-infected children and adults [74], making it worth considering as a vaccine antigen [20]. However, the antibody response against p27 may result in a selective pressure capable of inducing AA mutations in the p27 region of circulating hRSV strains. In this regard, Hause and colleagues [33] observed that subgroup B exhibited significantly more non-synonymous AA changes than subgroup A, documenting K124N, in accordance with our findings, as the most common AA change among hRSV-A [33].

On the contrary, in our study, hRSV-A strains were more frequently mutated in p27 than hRSV-B, although it should be noted that a different reference strain was herein considered.

Finally, it is worth noting the lack of mutated residues at site III of both hRSV subgroups and the unique AA change E463D found at an extremely low rate within site IV of hRSV-B, confirming the high evolutionary conservation of these key neutralizing epitopes due to the limited selective pressure in our geographic area [20,32,33,37,66,75], and thus, the low propensity to favor the occurrence of antibody-resistant viruses.

On the whole, AA changes causing a reduced or no binding affinity for the currently available mAbs [36,37,69] were not found in the hRSV Sicilian strains.

We also performed selective pressure analyses in the F protein genes of both viral subgroups. As a result, the F protein did not appear to be sensitive to the selective pressure, given that only one site under positive selection was estimated in the glycopeptide p27, according to the observations of other authors [76].

While most of studies have traditionally relied on the genetic variability of the two surface glycoproteins G and F, especially those focused on a phylogenetic analysis, the use of WGS data in evaluating the genomic heterogeneity of hRSV remains an exception. However, the higher resolution of this approach justifies its application in both surveillance studies and Public Health investigations, also considering the potential implications of different genomic regions on the preventive measures and treatments currently adopted in the community setting.

As also previously described, there are other genetic hotspots spanning the entire genome that carry several polymorphisms responsible for non-synonymous mutations in the other remaining structural or non-structural proteins.

Among the structural proteins, N and M showed the lowest variability, and low frequencies of non-synonymous AA changes were found in P, L and M2-1 proteins, thus confirming what has recently been proposed elsewhere [77,78]. Conversely, the small hydrophobic protein exhibited the highest proportion of mutated AA sites, suggesting a strong selective pressure on this genetic region [79], as similarly observed for the non-structural proteins M2-2 and NS2 in both hRSV-A and hRSV-B isolates [77,78,80].

There were some limitations in our study. First, this research was based on a single center, and this may have contributed to the introduction of a selection bias. Second, our sampling mostly included cases identified in the general population, and this may have caused the loss of relevant information. Third, the limited number of hRSV genomes did not allow for the comparison of mutational patterns per surveillance season.

Despite these limitations, our study provides a detailed picture of hRSV circulation patterns in Sicily, and to the best of our knowledge, this is the first study in Italy on the variability of hRSV whole-genome sequences collected over a wide time period encompassing six consecutive seasonal outbreaks, helping us further understand the characteristics of the global molecular epidemiology.

In conclusion, the low frequency of AA changes that we found, especially in the major antigenic sites targeting the currently licensed primary prevention treatments, which are likely not associated with a reduced susceptibility, may represent an advantage in the future use of mAbs and vaccines. However, we cannot exclude the possibility that additional sites may also be involved in the antibody response to hRSV, and novel AA variants in the virus, driven by these immunization programs yet unrecognized, might be of relevant interest in the future.

In this respect, this study emphasizes the need for global surveillance studies based on the characterization of hRSV whole-genome sequences, which may play a crucial role in tracking the evolution of contemporary strains and detecting the emergence of new genotypes and variants carrying AA variations, which may be able to interfere with the outcomes of prophylactic and therapeutic interventions.

## Figures and Tables

**Figure 1 viruses-16-00851-f001:**
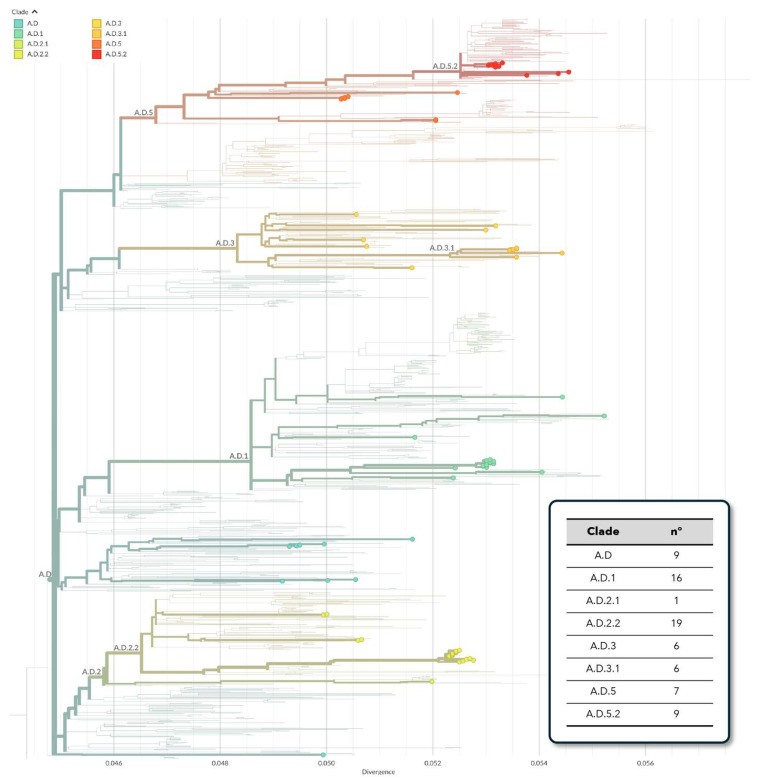
Clade assignment of Sicilian hRSV-A whole-genome sequences, as defined by Nextstrain.

**Figure 2 viruses-16-00851-f002:**
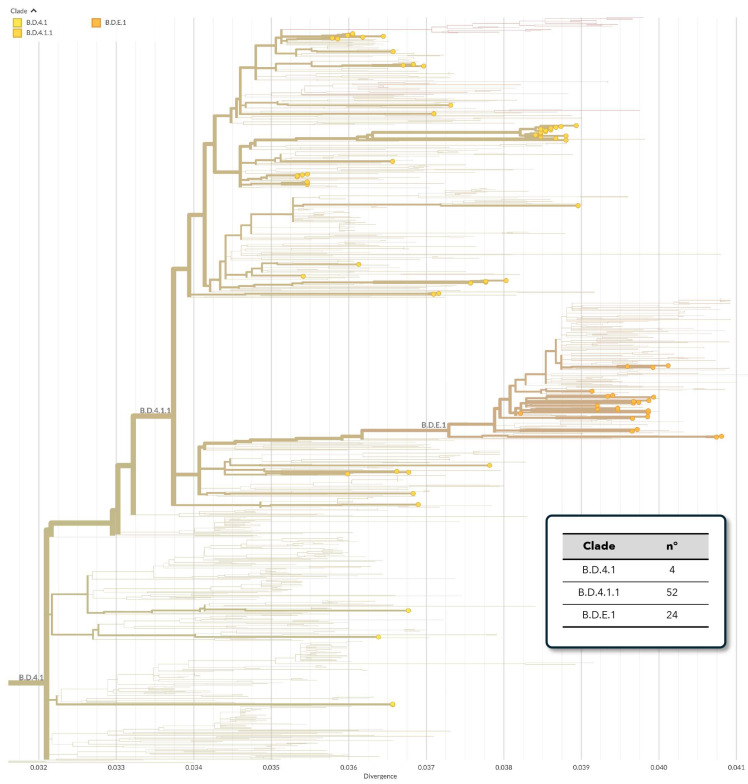
Clade assignment of Sicilian hRSV-B whole-genome sequences, as defined by Nextstrain.

**Figure 3 viruses-16-00851-f003:**
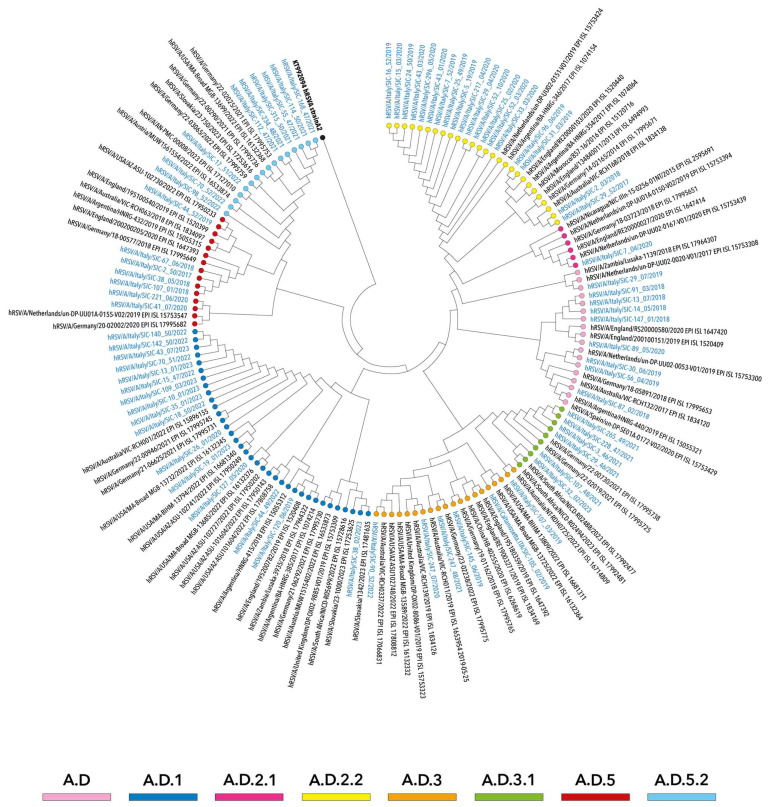
Neighbor-joining phylogenetic tree of whole-genome nucleotide sequences of hRSV-A strains collected in Sicily between October 2017 and April 2023. The study sequences are indicated in blue and by solid circles, differently colored according to the lineage. Reference sequences are indicated in black and are expressed in the following format: subgroup/country/strain/year–GISAID accession number. The strain A2 (GenBank: KT992094) was used as an outgroup (indicated as a black dot). Genotypes are defined according to the classification proposed by Goya S et al., 2020 [43].

**Figure 4 viruses-16-00851-f004:**
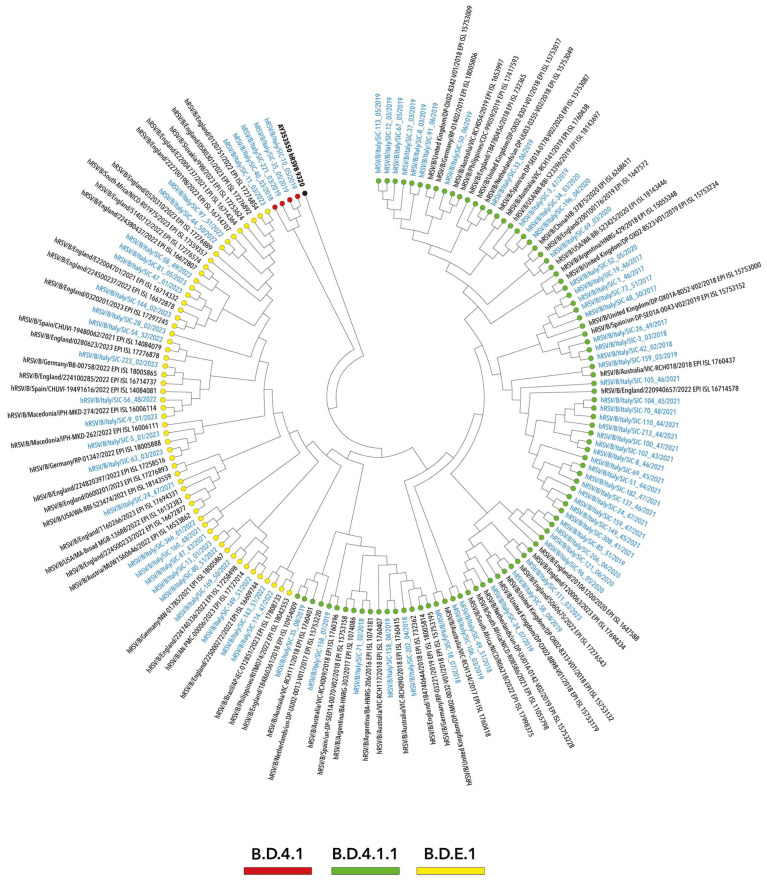
Neighbor-joining (NJ) phylogenetic tree of whole-genome nucleotide sequences of hRSV-B strains collected in Sicily between October 2017 and April 2023. The study sequences are indicated in blue and by solid circles, differently colored according to the lineage. Reference sequences are indicated in black and are expressed in the following format: subgroup/country/strain/year–GISAID accession number. The strain 9320 (GenBank: AY353550) was used as an outgroup (indicated as a black dot). Genotypes are defined according to the classification proposed by Goya S et al., 2020 [43].

**Figure 5 viruses-16-00851-f005:**
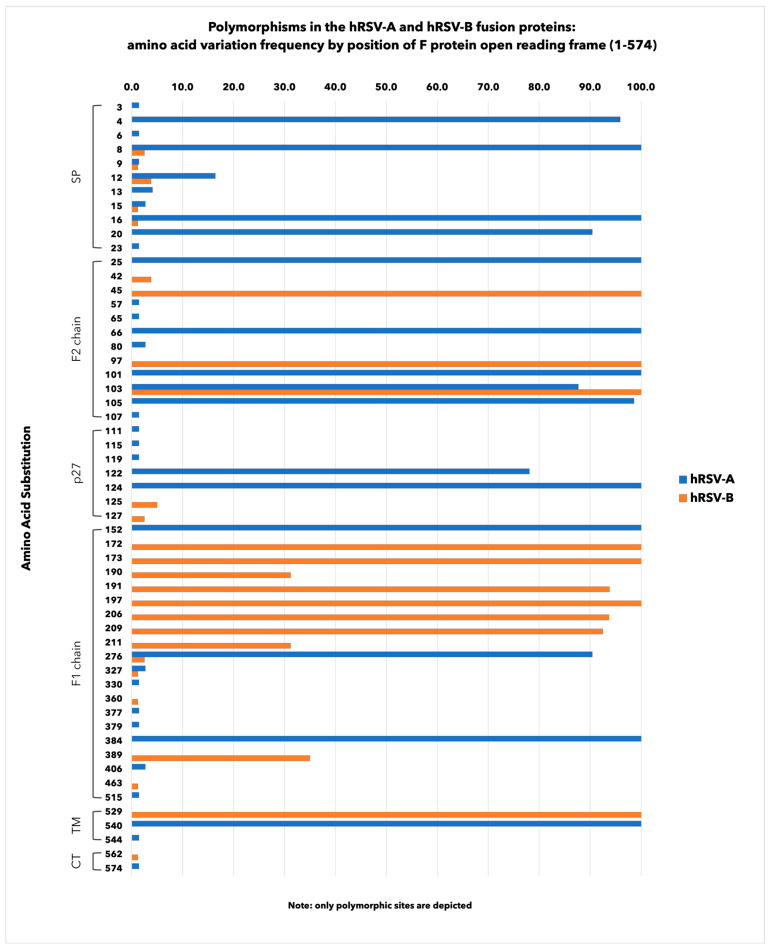
Amino acid variation frequencies detected in the F protein open reading frame of hRSV-A and hRSV-B Sicilian strains. Proportion of mutated AA positions: 6.3% for hRSV-A and 4.5% for hRSV-B.

**Table 1 viruses-16-00851-t001:** Demographic characteristics of the study population according to hRSV detection and subgroup. Period: 2017–2023 (% by row).

Demographic Characteristic	Total	hRSV Positive			
		Overall	hRSV-A	hRSV-B	hRSV-A + B
Study population [n (%)]	13,193	770 (5.8)	355 (46.1)	401 (52.1)	14 (1.8)
Age group [years; n (%)]					
≤11 months	356	81 (22.7)	36 (44.4)	43 (53.1)	2 (2.5)
12–23 months	588	135 (23.0)	65 (48.1)	66 (48.9)	4 (3.0)
2–4	1740	306 (17.6)	157 (51.3)	147 (48.0)	2 (0.7)
5–10	2000	108 (5.4)	50 (46.3)	52 (48.1)	6 (5.6)
11–18	954	22 (2.3)	11 (50.0)	11 (50.0)	0
19–34	1012	20 (2.0)	10 (50.0)	10 (50.0)	0
35–49	1518	11 (0.7)	4 (36.4)	7 (63.6)	0
50–64	2147	38 (1.8)	8 (21.0)	30 (79.0)	0
≥65	2878	49 (1.7)	14 (28.6)	35 (71.4)	0
Sex [n (%)]					
Female	6576	381 (5.8)	172 (45.1)	203 (53.3)	6 (1.6)
Male	6617	389 (5.9)	183 (47.0)	198 (50.9)	8 (2.1)
Healthcare setting [n (%)]					
Community based	7313	713 (9.8)	340 (47.7)	359 (50.3)	14 (2.0)
Hospital based	5880	57 (1.0)	15 (26.3)	42 (73.7)	0
Surveillance season [n (%)]					
2017–2018	2049	121 (5.9)	53 (43.8)	68 (56.2)	0
2018–2019	2209	170 (7.7)	36 (21.2)	131 (77.1)	3 (1.7)
2019–2020	2320	216 (9.3)	178 (82.4)	35 (16.2)	3 (1.4)
2020–2021	4	1 (25.0)	0	1 (100.0)	0
2021–2022	5284	107 (2.0)	30 (28.0)	75 (70.1)	2 (1.9)
2022–2023	1327	155 (11.7)	58 (37.4)	91 (58.7)	6 (3.9)

**Table 2 viruses-16-00851-t002:** Seasonal distribution of hRSV whole-genome sequences, according to hRSV subgroup. Period: 2017–2023 (% by row).

	Total	hRSV-A	hRSV-B
hRSV whole-genome sequences [*n* (%)]	153	73 (47.7)	80 (52.3)
Surveillance season			
2017–2018	22	11	11
2018–2019	28	9	19
2019–2020	32	23	9
2020–2021	1	0	1
2021–2022	31	12	19
2022–2023	39	18	21

**Table 3 viruses-16-00851-t003:** Amino acid substitution frequencies detected in the neutralizing epitopes Ø-V and in the internal glycopeptide p27 of the F protein of hRSV-A and hRSV-B Sicilian strains.

		hRSV-A		hRSV-B	
Neutralizing Epitope	Position	AA Substitution	%	AA Substitution	%
*Ø*	62–69 196–209	K65R	1.4	S197N	100.0
		K66E	100.0	I206M	93.7
				Q209R	92.5
*I*	27–45 312–318 380–400	V384I	100.0	R42K	3.8
				F45L	100.0
				S389F/P	35.0
*II*	254–277	N276S	90.4	S276N	2.5
*III*	46–54 305–310				
*IV*	422–471			E463D	1.2
*V*	55–61 146–194 287–300	I57V	1.4	S190N	31.2
		V152I	100.0	K191R	93.8
*Internal glycopeptide p27*	100–136	L111I	1.4	L125P/R	5.0
		M115I	1.4	V127A	2.5
		N119I	1.4		
		A122T	78.1		
		K124N	100.0		

## Data Availability

Sequence data are available on GISAID (https://gisaid.org, accessed on 26 May 2024).

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
