# Peer review of "Whole-Genome Sequencing and Genetic Diversity of Human Respiratory Syncytial Virus in Patients with Influenza-like Illness in Sicily (Italy) from 2017 to 2023"

_viruses, 2024, doi:10.3390/v16060851_

Round 1
Reviewer 1 Report
Comments and Suggestions for Authors
The manuscript entitled “Whole-genome sequencing and genetic diversity of human respiratory syncytial virus from patients with Influenza-like illness in Sicily (Italy), 2017 to 2023” by Tramuto F et al., is a study examining the sequence variability of the human respiratory syncytial virus. In particular, the authors collected a relatively high number of whole-genome sequences (N=153) of human respiratory syncytial virus proportionally distributed across six seasons (from 2017-2018 to 2022-2023). The topic of the paper is timely considering the aspect of human respiratory syncytial virus infection immunization. Overall, the paper is well written, the data presentation is clear, and only a few minor points should be addressed.
Comments:
Abstract
Ø In the abstract, the total number of patients included in the study and the total number of whole-genome sequences analyzed are missing. Please also add these data.
Results
Ø Paragraph 3.1, lines 251-252: It is not clear what is the total of the patients included in the analysis. Please, rephrase the sentence.
Ø Table 1: the % by row is not clear. It might be helpful to report the prevalence by column within each group. Moreover, the total of community-based and hospital-based participants within the RSV-A group (N=335) should be checked (N=340? and N=15?).
Ø Paragraph 3.2: Table S2 could be added to the main text because it would be helpful for the reader.
Ø RSV-A and B whole-genome sequences analysis: could an analysis that considers sites under positive or negative pressure, particularly for the F region in the two RSV genotypes, be added?
Comments on the Quality of English Language
Minor editing of English language required.
Author Response
We sincerely thank the Reviewers for providing us this opportunity to further revise our manuscript.
A point-by-point description of how each comment was addressed in the manuscript is given below. Original reviewers' comments in boldface, responses in regular typeface.
To Reviewer #1
- In the abstract, the total number of patients included in the study and the total number of whole-genome sequences analyzed are missing. Please also add these data.
Response
According to the Reviewer’s suggestion, in the abstract was reported the number of whole-genome sequences included in the study as well as the total number hRSV-positive subjects.
Results
- Paragraph 3.1, lines 251-252: It is not clear what is the total of the patients included in the analysis. Please, rephrase the sentence.
- Table 1: the % by row is not clear. It might be helpful to report the prevalence by column within each group. Moreover, the total of community-based and hospital-based participants within the RSV-A group (N=335) should be checked (N=340? and N=15?).
Response
We thank the reviewer for the suggestion. The sentence has been rephrased and a new column was added to the Table 1. Moreover, the total number of hRSV-A positive subjects was fixed.
- Paragraph 3.2: Table S2 could be added to the main text because it would be helpful for the reader.
Response
We fully agree with the Reviewer. The Table S2 has been added to the main text as Table 2 and the manuscript revised accordingly.
- RSV-A and B whole-genome sequences analysis: could an analysis that considers sites under positive or negative pressure, particularly for the F region in the two RSV genotypes, be added?
Response
In accordance with the Reviewer’s comment, the sites under selective pressure were evaluated in the F gene, applying the algorithms implemented in the Datamonkey webserver. Therefore, the paragraphs “2.4. Selection pressure prediction” in the Section: Materials and Methods, “3.4. Determination of the selection pressure on hRSV F gene” in the Section: Results and the Table S2 “Positive selection sites in F gene of hRSV-A and hRSV-B” have been added, as well as a new reference in the Discussion.
Reviewer 2 Report
Comments and Suggestions for Authors
The authors tried to genetic analyses based on full-genome sequences of the HRSV strains detected in Sicily during 2017-23. However, the authors missed many important items.
Major
The authors performed analyses using whole genome sequences. However, the authors could not make sense of genetic information.
1. The authors used NJ method. However, this is not suitable. At least, the author should use ML method and/or BMCMC method.
2. The authors made phylogenetic analyses based on the full-length nucleotide sequences. What dose it mean?
3. Although the authors did not analyze the G gene. The G gene analyses is useful. Why?
4. Introduction and Discussion are too long and described somethings unrelated to this study. Should improve them.
5. Figures were not sophisticated. Please improve them.
6. Format is not adjust for this journal. Should use suitable template for the journal after reading Instructions for the authors.  
Author Response
TO REVIEWERS
We sincerely thank the Reviewers for providing us this opportunity to further revise our manuscript.
A point-by-point description of how each comment was addressed in the manuscript is given below. Original reviewers' comments in boldface, responses in regular typeface.
To Reviewer #2
- The authors used NJ method. However, this is not suitable. At least, the author should use ML method and/or BMCMC method.
- The authors made phylogenetic analyses based on the full-length nucleotide sequences. What dose it mean?
Response:
We agree with the reviewer that the “Neighbor Joining method” is not the most accurate approach in reconstructing the phylogenetic relationships. However, it should be stressed that in this paper we aimed to provide an overall framework on hRSV evolution of full-length genomes.
We assume that the reviewer has suggested to address the phylogenetic analysis on the second hypervariable region of the G gene, as we carried out in a recent investigation [45].
As far as the scientific literature more commonly reports phylogenetic analyses based on the G gene, it is equally agreeable that the use of full-length genome sequences can only increase the phylogenetic resolution, as also suggested by several authors in different contexts.
Nevertheless, in our opinion, the present work mainly focused on the description of background polymorphisms due to the natural evolution, since active and passive primary prevention measures have not yet been implemented in Sicily.
- Although the authors did not analyze the G gene. The G gene analyses is useful. Why?
Response:
We agree with the reviewer that the analysis of the G gene is useful to understand the molecular variations of hRSV genomes. As reported in the manuscript (section Results: 3.4. Analysis of NS1, NS2, N, P, SH, M2-1, M2-2, and L proteins), the G protein was not considered in the present study because it was already discussed by our research group in a previous paper [45].
- Introduction and Discussion are too long and described somethings unrelated to this study. Should improve them.
Response:
According to the reviewer’s comment, the entire manuscript was revised prioritizing the topics related to to this study.
- Figures were not sophisticated. Please improve them.
Response:
We sincerely apologize but we do not understand what the reviewer means with “not sophisticated”. We will be pleased to welcome any request that the reviewer will provide.
- Format is not adjust for this journal. Should use suitable template for the journal after reading Instructions for the authors.
Response:
We understand the reviewer’s concern and we agree with him. However, we have followed the “Instruction for the authors --> Manuscript Submission Overview” where is stated that a “Free Format Submission” is accepted by Viruses MDPI.
Anyway, thanks to the welcome support of the editorial office, the manuscript is now formatted on the Viruses Microsoft Word template.
Reviewer 3 Report
Comments and Suggestions for Authors
Please find the attached file.

My suggestions are included in the above file.
Author Response
TO REVIEWERS
We sincerely thank the Reviewers for providing us this opportunity to further revise our manuscript.
A point-by-point description of how each comment was addressed in the manuscript is given below. Original reviewers' comments in boldface, responses in regular typeface.
To Reviewer #3
- For the six seasons, the predominant subgroups were shown as “BBABBB”. How important is it to define the yearly prevalence patterns this way? Is it a part of the worldwide patterns? Does it provide any information in terms of future epidemiology of RSV?
Response:
The epidemiology of hRSV in Sicily was previously documented by our research group, considering Pre- and Post-COVID-19 Surveillance Seasons [45]. The present paper focused on the genetic diversity of hRSV genomes obtained through next-generation sequencing (NGS), then, we preferred to report the most prevalent hRSV subgroup in a concise manner. Nevertheless, we are available to revise the manuscript whether the reviewer should consider it to be inappropriate this way.
- During 2020-2021 season, only 1 positive sample was sequenced. Is this because a small number of samples were collected due to the pandemic restriction? Is it because RSV prevalence was reduced due to less human-to-human interaction? Or both? It may be useful to show the total number of specimens subjected each year (out of 13,221 specimens) in Table 1.
Response:
We thank the Reviewer for the comment. In 2020-2021, due to the strong impact of COVID-19 pandemic, a disruption of the national ILI surveillance system occurred. Moreover, a reduction in the circulation of hRSV was observed, because of the widespread implementation of non-pharmaceutical preventive measures [45]. As a consequence, the number of collected samples was nearly negligible and only one hRSV-positive sample was detected.
Table 1 was revised according to the reviewer’s suggestion. A new column reporting the total number of samples (and percentage) by category was added.
Additional suggestions:
- On page 3,
none of the polymorphisms know -> none of the polymorphisms known
- On page 5,
with the aim to passive protect -> with the aim to passively protect
- On page 5,
For these reasons, as seen with influenza, continued molecular surveillance of hRSV is essential to investigate the epidemiology of the virus and the circulation of subgroups, but it is also an invaluable tool for understanding ->
For these reasons, as seen with influenza, continued molecular surveillance of hRSV is essential to investigate the epidemiology of the virus and the circulation of subgroups; it is also an invaluable tool for understanding
(I think the whole sentence is too long. Both the first and second part of the sentence emphasize the importance of molecular surveillance of hRSV. Hence, they can be conjunctive using a semicolon.)
- On page 7,
ILI or SARI
(The full names of ILI and SARI can be indicated where these words appear for the first time.)
- On page 11,
The male-to-female ratio was 1.02 and, also, no differences were found -> The male-to- female ratio was 1.02 and no difference was found
- On page 11,
in B.D.E.1 felt a small number of sequences more recently collected between 2021-2022 and 2022-2023
(I could not understand this sentence. Does it mean “a small number of sequences more recently collected between 2021-2022 and 2022-2023 fall into B.D.E.1”?)
- Figure 2 (inset legend),
D.B.E.1 -> B.D.E.1
- On page 17,
two of which A122T and K124N -> two of which were A122T and K124N
- On page 20,
However, strategies currently adopted in active and passive primary prevention of hRSV infections target different surface glycoproteins, among which the F protein demonstrated the most promising results, because of unique properties involving its role in cell entry, the higher genetic conservation and the presence of antigenic epitopes, in the pre-fusion conformation, which are able to bind antibodies with the strongest neutralizing activity. ->
However, strategies currently adopted in active and passive primary prevention of hRSV infections target different surface glycoproteins. Among them, the F protein demonstrated the most promising results because of its unique properties involving its role in cell entry, the higher genetic conservation and the presence of antigenic epitopes in the pre-fusion conformation where antibodies may bind with the strongest neutralizing activity.
(This is an example of reorganization, since I thought the whole sentence is too long and complicated.)
- On page 20,
Because their quasispecies nature, RNA viruses, including hRSV, undergo constant evolution. Therefore, immune escape mutants may emerge as consequence of an adaptation ->
Because of their quasispecies nature, RNA viruses, including hRSV, undergo constant evolution. Therefore, immune escape mutants may emerge as a consequence of adaptation
- On page 22,
However, it cannot be excluded the possibility -> However, we cannot exclude the possibility
Response:
We sincerely appreciated all the helpful tips, which have been accepted in their entirety. The manuscript has been revised accordingly.
Round 2
Reviewer 2 Report
Comments and Suggestions for Authors
The authors well addressed for my comments, excepting phylogenetic construction method. However, this is a important paper regarding local molecular epidemiology in Italy. Thus, I recommended that this was acceptable for publication in Viruses.